# Extreme Body Condition Index Values in Small Mammals

**DOI:** 10.3390/life14081028

**Published:** 2024-08-19

**Authors:** Linas Balčiauskas, Laima Balčiauskienė

**Affiliations:** Nature Research Centre, Akademijos 2, 08412 Vilnius, Lithuania; linas.balciauskas@gamtc.lt

**Keywords:** mice, voles, shrews, body condition extremes, fitness, Chitty effect

## Abstract

The body condition index (BCI) values in small mammals are important in understanding their survival and reproduction. The upper values could be related to the Chitty effect (presence of very heavy individuals), while the minimum ones are little known. In this study, we analyzed extremes of BCI in 12 small mammal species, snap-trapped in Lithuania between 1980 and 2023, with respect to species, animal age, sex, and participation in reproduction. The proportion of small mammals with extreme body condition indices was negligible (1.33% with a BCI < 2 and 0.52% with a BCI > 5) when considering the total number of individuals processed (*n* = 27,073). When compared to the expected proportions, insectivores and herbivores were overrepresented, while granivores and omnivores were underrepresented among underfit animals. The proportions of granivores and insectivores were higher, while those of omnivores and herbivores were lower than expected in overfit animals. In several species, the proportions of age groups in underfit and overfit individuals differed from that expected. The male–female ratio was not expressed, with the exception of *Sorex araneus*. The highest proportion of overfit and absence of underfit individuals was found in *Micromys minutus*. The observation that individuals with the highest body mass are not among those with the highest BCI contributes to the interpretation of the Chitty effect. For the first time in mid-latitudes, we report individuals of very high body mass in three shrew species.

## 1. Introduction

The three universal ecological responses to global warming are proposed and tested: shifts of species distributions [1], changes of phenology in plants and animals [2], and reduction in body size [3]. The distribution shifts have been quantified: the median rate of the shift to higher latitudes is 16.9 km per decade [1]. However, the details of the changes in body size are not yet known [4]. The magnitude of body size shrinkage differs across species [5]; therefore, heterogeneity of responses might have unpredictable consequences for ecosystems [4].

Mammals exhibit a wide range of body size responses to climate changes, yet the underlying mechanisms remain poorly understood. Changes in the principal size components can be related to both body length and body mass, as evidenced by Li et al. [6]. Consequently, comparisons between different species and populations are often challenging. In our understanding, the body condition index based on these two size measures [7] can reflect both changes equally [8].

In contrast, some mammals are capable of increasing their body size [9], a phenomenon that is contingent upon a number of factors, including the availability of resources, the passage of time, and the geographical context, which encompasses factors such as latitude, the presence of an island habitat, and other environmental variables [10]. Temporal changes in body size can differ due to life history traits, such as reproductive rate, and habitat. However, investigations of these factors have been limited [11].

In the context of global climate change, the evolution of wildlife is an unavoidable phenomenon [12]. To survive, species can either shift their ranges or adapt. Adaptations can be observed at the level of animal ecology, behavior, and physiology, all of which are related to body size [13]. The body size of a species tends to increase in a latitudinal and elevational gradient, a phenomenon known as Bergmann’s rule [14]. This phenomenon is more pronounced in small mammals, such as the herb field mouse (*Apodemus uralensis*) [15], than in shrews [16].

The impact of disturbances, such as urbanization, on animal body size and fitness is less understood. However, it is likely that these disturbances will affect animal fitness due to increased temperatures, habitat changes and resource availability [17]. For rodents, it has been demonstrated that climate change and human-caused disturbance have a simultaneous influence on population decline in various ecosystems [18].

An understanding of the fitness of animals provides a more comprehensive knowledge about the ecosystems in which they live. Body condition is related to reproduction [19]. However, intensive reproduction has benefits for population growth but also imposes costs on the individual [8]. Furthermore, better body condition leads to better survival, enabling animals to overcome weather constraints [20]. However, lower body mass, which is reflected in a lower body condition index, results in better overwintering in several vole species, as evidenced by studies by Ergon et al. [21], Balčiauskienė et al. [22], and Zub et al. [23]. Fitness can be related to the survival of species [24], which has implications for species conservation and habitat management [25]. Besides, body condition of small mammals can be used as an indicator of environmental pollution [26,27,28].

Among the various methods employed to assess the fitness of animals, there are those based on:Morphological measurements, such as body mass, body length, and indices based on various other traits;Physiological indicators, such as hormone levels or blood parameters;Reproductive success, including litter size or breeding frequency;Survival, including mortality and longevity;Immunology, including the measurement of immune responses or pathogen loads.

These methods are species-specific and are described in greater detail in [25].

The assessment of fitness in small mammals is a topic of ongoing debate, with various body condition indices (BCIs) being employed to this end. The optimal BCI remains a matter of contention [29]. The use of body mass as a proxy for fitness in wild animals is limited due to the incomplete understanding of the relationship between the two variables [30]. Various indices calculated as a ratio of body mass to body size have been found to function better, but they are still not without critics [31]. Additionally, estimation of fat amount—either visually, through dissection, or using ultrasound and other techniques—is employed [24], as is the estimation of lean body mass [32]. While we agree with the aforementioned authors, the proposed measures have two shortcomings. Firstly, some of them necessitate the use of specific laboratory equipment [33], and secondly, there is a lack of retrospective data for their comparisons.

Given the natural variability of size in mammals [34], it is to be expected that extreme values of body mass/BCI will be observed in small mammals. While the presence of overweight individuals is a component of the Chitty effect [35,36], the low body mass of various small mammal species remains outside the scope of this analysis, with the exception of the analysis conducted by R. Boonstra and C.J. Krebs [37]. The researchers observed differences in survival between large and small individuals of four species of voles. However, the range of body mass classified as “small” was quite broad.

Our interest in the BCI extremes in small mammals was also prompted by the observation that a greater proportion of high-cited investigations relate to the higher latitudes [38,39,40,41,42] than to mid-latitudes [43,44,45,46]. Furthermore, the majority of studies have focused on a limited number of species, namely gray voles and shrews, with minimal sampling of other small mammal species in a given area. Additionally, the Chitty effect has been reported as absent in shrews, as evidenced by Nordahl and Korpimäki [41], yet it has been observed in Lithuania [8].

The objective of this study was to examine the distribution of small mammals exhibiting either extremely high (referred to as the Chitty effect) or low body mass with respect to their trophic group, species, sex, age, and reproductive intensity. Additionally, we sought to estimate the relationship between body condition index and the mass of individuals of different species required for the Chitty effect to occur.

## 2. Materials and Methods

### 2.1. Study Site, Small Mammal Collection Methods, and Sample Size

For this study, we subsampled data on small mammals trapped in Lithuania between 1980 and 2023. The climate of Lithuania is generally classified as humid continental, with some regional variations influenced by proximity to the Baltic Sea. However, Lithuania does not have distinctly separate climatic zones.

Most of the country’s 65,286 square kilometers is flat land. The dominant landscape, over 75% of Lithuania, is rural. In 2012, about 33% of the territory was covered by forests, 33% by arable land and permanent crops, and 27% by semi-natural vegetation. Land cover in the country changes at a rate of about 0.48% per year. About 15% of the territory is covered by natural landscapes [47]. The Lithuanian climate is mild. For the period 1991–2020, the average annual air temperature in Lithuania was 7.4 °C. The warmest month of the year is July (18.3 °C) and the coldest is January (–2.9 °C). The average annual rainfall in Lithuania is 695 mm. The highest precipitation, 84 mm, occurs in July and the lowest, 37 mm, in April [48]. Although we do not exclude the possible influence of climate warming on body mass loss and BCI, the lack of data from different latitudes does not allow us to exclude the possible occurrence of Bergmann’s rule [15], so the data are not analyzed in temperature context.

We included only species with sample sizes of *N* > 50, for which all required measurements (body mass and body length) were available to calculate the body condition index. Therefore, we analyzed 12 small mammal species (Table 1). Due to the subsampling methodology employed, the resulting numbers and proportions differ from those reported in [8,49].

The primary method employed for the collection of specimens was the use of snap traps. These traps were arranged in lines of 25 traps, 5 m apart. One or more such lines were set for three days, with checks conducted once a day in the evening or twice a day in the morning and evening. The use of alternative methods, such as live traps and pitfall traps, resulted in the capture of less than 0.01% of individuals.

Most of the small mammals (76.1%) were caught in the fall season, 13.3% in the summer, 6.3% in the spring and 4.3% in the winter. However, season was not a strong factor (*η^2^* = 0.021), nor was animal age (*η^2^* = 0.013), being much weaker than species and habitat [49].

### 2.2. Laboratory Processing of Captured Small Mammals

If the material was not processed immediately after capture, the captured individuals were stored in a frozen state until processing. Their species were identified based on external features; *Microtus* voles were identified by their teeth [50].

Prior to dissection, the weight of each captured small mammal was recorded to the nearest 0.1 g, and their body length, tail length, hind foot length, and ear length were recorded to the nearest 0.1 mm using calipers. To minimize potential measurement bias, as described by Krebs and Singleton [7], more than 80% of the measurements were conducted by the same individual throughout the study period.

This study defined three age groups in small mammals: adults (including all reproducing and post-reproductive individuals), subadults (with developed but not active genitalia), and juveniles (genitalia not developed). Additionally, the presence or atrophy of the thymus gland was tested. The thymus gland is fully involuted in adults, partially involuted in subadults, and well-developed in juveniles, as described by R.C. Terman [51]. In instances where gonadal status and thymus were unavailable, body mass was employed as a proxy. A comprehensive account of the methodology employed for the identification of age groups can be found in [8]. Observed proportions of age and sex groups in the analyzed species are presented as Table A1; observed numbers were reported in [8].

In adult males, five levels of reproduction intensity were defined according to [52]: No spermatogenesis (0)—thymus absent, scrotal testes;Beginning of spermatogenesis (1+)—enlarged testes but empty epididymis;Medium intensity spermatogenesis (2+)—enlarged testes and epididymis, empty or slightly enlarged seminal vesicles;Spermatogenesis intensive (3+)—full epididymis and seminal vesicles;Slumped and slate colored testes and epididymis, plus empty seminal vesicles characterizing males after reproduction.

In adult females, the number of litters was identified based on the signs listed in [53,54], which included perforated or plugged vagina, lactating, pregnancy, and post-pregnancy (*corpora lutea* and placental scars). No more than three litters can be identified by combining the listed traits [8].

### 2.3. Data Analysis

The body condition index (BCI) was calculated according to the formula proposed by Moors: BCI = (Q/L^3^) × 10^5^, where Q is the body weight in grams, and L is the body length in millimeters [55]. For pregnant females, the weight of the embryos was excluded [56]. The relationship between body length and body mass was examined in all analyzed species, and a strong correlation was identified in rodents (Pearson’s r between 0.79 and 0.90, all *p* < 0.0001). In shrews, the correlations were weaker, yet still significant: *S. araneus* (r = 0.50), *S. minutus* (r = 0.29), and *N. fodiens* (r = 0.55), all with *p*-values less than 0.001. In all species, the relationship was linear, as required by [31] regarding the appropriate use of body condition indices.

In Lithuania, the mean BCI for all small mammal species, excluding rats, was 3.03, with a minimum of 1.04 and a maximum of 6.89. In all species, irrespective of their BCI mean, mean minus standard deviation values were between 2.13 and 2.94, while mean plus standard deviation values were below 4 in all species except *M. musculus* and *M. minutus,* with mean + SD = 4.16 and 4.49, respectively [8]. Based on this, the body condition index extremes were defined as BCI < 2.0 and BCI > 4.0, with a BCI > 5.0 considered exceptionally large. The distribution of maximally and minimally fit individuals by species, age, sex, and reproduction intensity was examined. Proportions were expressed as percentages. The observed proportions were tested for differences using the chi-squared test, with the theoretical assumption that the expected proportion of overweight or underweight individuals should correspond to the sample composition of the species. For the chi-squared test, we employed the “sample vs. expected” approach with Monte-Carlo permutation (*N* = 9999) in PAST, version 4.13 (Museum of Paleontology, Oslo College, Oslo, Norway) [57]. To assess the reliability of differences in BCI between individuals with the highest body weight and all adult individuals, we conducted a Student *t*-test. We set the minimum confidence level in all tests at *p* < 0.05.

## 3. Results

### 3.1. Body Condition Index Extremes in Small Mammal Species and Trophic Groups

A total of 361 individual small mammals belonging to 11 species exhibited a BCI below 2.0, representing 1.3% of the total number (Table 2). The observed proportions of individuals with low BCI were not in accordance with the number of analyzed individuals of different species (Figure 1a; χ^2^ = 181.8, *p* < 0.0001). The proportion of individuals with low BCI was found to be higher than expected in *S. minutus* (χ^2^ = 226.5, *p* < 0.0001), *S. araneus* (χ^2^ = 146.8, *p* < 0.0001), and *M. arvalis* (χ^2^ = 8.3, *p* < 0.01). The proportions were found to be lower than expected in *A. flavicollis* (χ^2^ = 57.4, *p* < 0.0001), *A. agrarius* (χ^2^ = 31.0, *p* < 0.0001), *C. glareolus* (χ^2^ = 9.9, *p* < 0.01), *and M. musculus* (χ^2^ = 4.2, *p* = 0.05). In other species, the proportions of individuals with a BCI < 2 were found to be in accordance with their representation in the sample (Table 2). No underweight individuals were found in *M. minutus*.

A total of 1502 individuals exhibited BCI > 4, with observed proportions differing from those expected (Figure 1a, χ^2^ = 194.5, *p* < 0.0001). In two species, *M. musculus* and *M. minutus*, the proportion of overweight individuals was 3–6 times greater than expected (χ^2^ = 105.0 and 234.4, respectively, *p* < 0.0001). In *S. minutus* (χ^2^ = 4.3, *p* < 0.05), *A. agrarius* (χ^2^ = 24.1, *p* < 0.0001), and *A. flavicollis* (χ^2^ = 23.3, *p* < 0.0001), the proportion of overweight individuals was approximately 1.4 times higher than expected (Table 2).

The proportion of individuals with a BCI > 4 in voles, *C. glareolus*, was less than expected (χ^2^ = 87.5, *p* < 0.0001), as was the proportion in *M. arvalis* (χ^2^ = 7.6, *p* < 0.01), *M. agrestis* (χ^2^ = 6.9, *p* < 0.01), and *A. oeconomus* (χ^2^ = 4.4, *p* < 0.05).

A total of 140 individuals (0.52%) exhibited BCI values greater than 5, which deviated from the expected proportions (Figure 1a, χ^2^ = 56.0, *p* < 0.0001). The highest proportion of overweight individuals was observed in *M. musculus* and *M. minutus*, with a prevalence that was 5.4–13.6 times greater than expected (χ^2^ = 39.3 and 226.7, respectively; *p* < 0.0001). The other species with a higher than expected proportion of BCI > 5 were *S. minutus* (χ^2^ = 12.9, *p* < 0.001) and *A. oeconomus* (χ^2^ = 3.9, *p* < 0.05). In contrast, the proportion of overweight individuals in *C. glareolus* (χ^2^ = 22.9, *p* < 0.0001) and *M. arvalis* (χ^2^ = 5.7, *p* < 0.02) was less than expected (Figure 1a, Table 2).

The proportion of individuals with low or high BCI did not correspond to the number of individuals analyzed from different trophic groups (Figure 1b). So, among individuals with a BCI < 2, there were four times more insectivores and 1.4 times more herbivores than expected, but nearly eight times fewer granivores and 1.5 times fewer omnivores (χ^2^ = 530.4, *p* < 0.0001).

Among the individuals with high BCI, the differences were less expressed. Among the individuals with a BCI > 4, there were 1.5 times more granivores, 1.5 times fewer omnivores, and 1.4 times fewer herbivores, while the number of insectivores was approximately the same as expected (χ^2^ = 202.1, *p* < 0.0001). Among individuals with a BCI > 5, there were 1.5 times more granivores and 1.4 times more insectivores than expected, but the number of overweight omnivores was only half that expected, and the number of herbivores was 1.25 times less (χ^2^ = 28.7, *p* < 0.0001).

### 3.2. Sex- and Age-Related Issues in Body Condition Index Extremes

A consistent pattern of sex-related variation in extremes of body condition index was not identified. Furthermore, the number of significant differences was limited (Figure 2a). Among individuals with a BCI < 2, the proportion of females (1:1.75) was higher than expected only in *S. araneus* (χ^2^ = 6.2, *p* < 0.05).

Among individuals with a BCI > 4, the proportion of females was higher than expected in *A. agrarius* (1:1.15, χ^2^ = 6.2, *p* < 0.05) and *A. oeconomus* (1:14.5, χ^2^ = 13.7, *p* < 0.0002). Additionally, a trend was observed in *C. glareolus* (1:1.20, χ^2^ = 2.9, *p* < 0.10), although it did not reach statistical significance. In the remaining species, the observed proportions were consistent with the expected sex ratio for the entire species sample.

The observed male-to-female ratio among individuals with a BCI > 5 did not differ from the expected ratio (Figure 2a).

Inconsistency with expected age proportions was observed in five small mammal species with a BCI < 2 (Figure 2b). The number of low-fitted juveniles was found to be higher than expected in *S. araneus* (χ^2^ = 17.3, *p* < 0.001), *S. minutus* (χ^2^ = 5.4, *p* < 0.01), and *C. glareolus* (χ^2^ = 15.1, *p* < 0.001). In *A. oeconomus*, adults were overrepresented, while subadults and juveniles were underrepresented (χ^2^ = 6.5, *p* < 0.05). In *M. arvalis*, subadults were overrepresented, while adults were underrepresented compared to expected numbers, though the difference was on the trend level only (χ^2^ = 4.6, *p* = 0.10).

Among individuals with a BCI > 4, there was a notable overrepresentation of adults and an underrepresentation of juveniles in *S. araneus* (χ^2^ = 11.7, *p* < 0.005). Among individuals with a BCI > 5, juveniles of *M. musculus* and *A. agrarius* were overrepresented, while the proportion of adults in both species was lower than expected (χ^2^ = 7.2 and 6.7, respectively, *p* < 0.05 in both species).

### 3.3. Reproduction-Related Issues in Body Condition Index Extremes

The occurrence of extreme values of BCI was less prevalent in reproducing individuals, both males and females. The frequency of reproductive males with a BCI < 2 was only 0.3%, while that of females was 1.2% (Table 3). The frequency of reproductive males with a BCI > 5 was 0.1%, while that of females was 0.3%. These figures are less than those observed in general (see Table 2). No reliable statistical differences were found depending on the intensity of reproduction.

### 3.4. Body Mass of Individuals with Highest Body Condition Indices

A total of 123 small mammals of known age were identified in the sample, with a BCI > 5. Of these, 61 were juveniles, 33 were subadults, and 29 were adults (Table 4). The distribution of age groups across species was similar, with some exceptions.

No juveniles were present among individuals characterized by BCI > 5 in *S. araneus*, *S. minutus*, and *M. agrestis*, no subadults were present in *M. arvalis* and *A. oeconomus*, and no adult individuals were present in *A. agrarius* and *M. arvalis*. These species are representative of insectivores, granivores, and herbivores.

The range of body masses producing BCI values greater than 5 was likely to be quite broad, with a ratio between the minimum and maximum values being less than 2 in shrews, *M. musculus*, *M. minutus*, and *C. glareolus* (Table 4). This ratio was generally higher in granivores and herbivores, with a maximum in *A. flavicollis* of 2.5 in juveniles and 3.3 in adults, and in *A. oeconomus,* 2.6 in juveniles and 2.4 in adults.

### 3.5. Chitty Effect? Maximum Body Mass and BCI of Adult Individuals

Our findings indicate that the highest BCI values in small mammals were not correlated with body mass. In other words, the heaviest individuals were not necessarily characterized by the highest BCI (Table 5). The threshold body mass for each species was arbitrarily set. Given the absence of differences in the male-to-female proportion among the heaviest individuals, we did not present the sex ratio.

The highest proportions of the heaviest individuals, exceeding 5% of the total number of adult individuals, were observed in *M. minutus* and *N. fodiens*. In contrast, in *C. glareolus*, *A. flavicollis*, and *M. arvalis*, these proportions were less than 1% (Table 5).

In only two species, *N. fodiens* and *M. agrestis*, the maximum observed BCI was observed in one of the heaviest individuals.

The heaviest individuals exhibited BCI values that were significantly higher than those observed in all adult individuals of *C. glareolus* (*p* < 0.01), as well as in *S. araneus*, *S. minutus*, *M. minutus*, and *M. arvalis* (*p* < 0.05). In other species, these differences were not found to be statistically significant.

## 4. Discussion

The results demonstrate that only a small proportion of individuals exhibited extreme body condition indices. Those with a BCI < 2 comprised 1.33%, and those with a BCI > 5 just 0.52% of the analyzed small mammals. Among individuals with a BCI < 2, the proportion of insectivores and herbivores was higher, while that of granivores and omnivores was less than expected. Among individuals with a BCI > 5, the proportion of granivores and insectivores was higher, while that of omnivores and herbivores was less than expected.

The average proportion of extra-large adults was 1.0%, with the highest proportion observed in *M. minutus* and *N. fodiens* (6.9% and 6.3%, respectively). The least represented large adults (>1%) were in *C. glareolus*, *A. flavicollis*, and *M. arvalis*. This finding is compared to data from other populations.

The absence of extra-large individuals in *S. araneus* was reported for Finland [41]. Furthermore, the loss of cyclic fluctuations, which has resulted in the disappearance of the Chitty effect in eight shrew species, particularly those with non-cyclic population dynamics, was reported from the Middle Yenisey at high latitude [58,59]. Consequently, our report of the maximum body mass observed in three shrew species from Lithuania represents a novel contribution to the existing body of knowledge.

In the California vole (*Microtus californicus*), the proportion of extra-large individuals was 12.7% [44]. In the meadow vole (*Microtus pennsylvanicus*), the proportion extra-large individuals during a three-year period was 0.47–0.77% [60].

A mere 0.06% of *C. glareolus* individuals with a body mass of 33.5–35.0 g was observed in the Białowieża National Park, Poland, a region with a similar latitude to Lithuania [61]. The proportion of overweight *C. glareolus* in our sample was significantly higher (χ2 = 14.8, *p* < 0.001) at 0.4%, despite the use of a minimum body mass of 35.0 g to select for this category (see Table 5). In the Middle Ural region, *C. glareolus* with body mass over 35.0 g comprised 0.98% of all individuals [42], which is a significantly (χ^2^ = 7.0, *p* < 0.01) higher proportion than in our sample, this being 0.38%.

In a study conducted in high latitudes by J. Tast in 1972, no individuals of the species *A. oeconomus* were reported to exceed the size of an extra-large individual [62]. However, in an earlier study by the same author, 6.5–10.0% of males were reported to be very large, exceeding 80 g [63]. In our study, the maximum body mass of this species was 77.0 g, which is comparable to the body mass reported for Germany by G.H.W. Stein in 1952 [64].

D. Chitty [38] reported no individuals of *M. agrestis* with a body mass exceeding 40.2 g for males and 36.2 g for females. However, the minimum body mass of adult females (18.0 g) and males (23.0 g) was comparable to that observed in our sample. We found a total 15 out of 301 adult *M. agrestis* to be of low weight, with males weighing between 18 and 21.5 g and females weighing between 16.5 and 21.3 g. These figures are somewhat lower than those reported in [38].

Similarly to our findings, the minimum and maximum body masses of adult *A. agrarius* from Slovakia were reported to be 14.5–44.0 for males and 14.0–46.0 for females [65]. However, the proportion of these large individuals in the sample was not presented.

The viability of large-sized Townsend’s vole (*Microtus townsendii*), prairie vole (*Microtus ochrogaster*), *M. pennsylvanicus*, and *M. californicus* was demonstrated in [37], yet the authors do not present the numbers of these individuals. Another paper, which deals with body mass of six small mammal species, does not present data that can be used in our context [66]. Therefore, we believe that the reference data provided are valuable for comparison with other territories, as well as species that have not been analyzed in this respect. In our dataset, the issue of measurement bias, which was highlighted in [7,66], has been effectively addressed.

The proportions of underfit individuals exhibited significant differences from those expected in *S. araneus*, *S. minutus*, *A. flavicollis*, *M. arvalis*, *A. agrarius*, *C. glareolus*, and *M. musculus*. Additionally, the proportions of overfit individuals in the last four species differed from those expected.

In our sample, *M. minutus* was distinguished by the absence of underfit individuals and the highest proportion of overfit individuals. In Austria, W. Haberl and B. Kryštufek reported the maximum body mass of adult females as 12.0 g, and that of adult males as 10.6 g [67]. In the high-latitude site in Finland, the maximum adult body mass was 13.5 g in females and 9.0 g in males [68]. In our sample, adult females with a body mass > 12.0 g comprised 10%, while the maximum body mass in males was 13.5 g.

The observed male-to-female ratio among individuals with a BCI > 5 and those with a BCI < 2 did not differ from the expected ratio in our sample; the proportion of females was higher than expected only in *S. araneus*. This can be explained by the life history of this species, which involves short age and large litters at considerable energetic costs [69,70]. Consequently, the reproductive cost represents a trade-off between fitness and body condition [19].

Additionally, our findings revealed that the age groups of certain small mammal species with extreme BCI values did not adhere to the expected distribution. In comparison to the anticipated ratio, the number of juveniles was higher than expected in *S. araneus*, *S. minutus*, and *C. glareolus*, while the number of subadults was disproportionately represented in *M. arvalis*, and the number of adults was overrepresented in *A. oeconomus*.

With regard to shrews, an explanation may be sought in their metabolic rate, which is high in shrews in general and particularly high in the smallest ones (juveniles). This is coupled with the specificity of their diet, which contains 100% of the high-quality food [71]. It is also possible that differences in the diet of *C. glareolus* juveniles may be expected in connection with their use of lower-quality habitats [72]. The most promising explanation for the low BCI in subadults is that they experience stress and competition at postnatal dispersion [73,74]. Low body mass *A. oeconomus* are characterized by better survival in the beginning of winter [23]; moreover, this species does not tolerate even short-term food deprivation [75].

Our results indicated that the proportion of juvenile *M. musculus* and *A. agrarius* among individuals with a BCI > 5 was higher, while the proportion of adults was lower than expected for both species. The high body condition in juveniles may be related to a lower pathogen load, which allows for more energy to be utilized for growth [76], and possibly to maternal care. It can be postulated that species with high adaptability and potential for spread should exhibit better body condition in juveniles and adults. This is evidenced by the fact that both *M. musculus*, which occupies a wide range of habitats [77], and *A. agrarius*, which has recently been documented to have a wide distribution [78,79], exhibit this characteristic. However, data on the body condition of these species from different regions within their ranges are currently unavailable.

Consequently, we posit that the comparison of the extremes of body condition for the most abundant small mammal species in a single region, employing a methodology that permits the re-recruitment of retrospective data from other sites, represents a significant advantage of this publication.

It is important to acknowledge the limitations of this study. Firstly, it is not possible to retrospectively assess the nutritional condition of individuals. Our BCI was traditional (based on morphometry); therefore it was not possible to correctly assess the nutritional condition of an individual due to the dynamics of body tissues [80]. It was not possible to perform retrospective measurements of body composition, as in [81], and thus we relied on the assumption that higher BCI values were beneficial to the individual. Secondly, we did not relate BCI values to population cycles across species. In Lithuania, population cycles are not a mandatory attribute of small mammal species [82]. This is also true of other territories outside high latitudes [83]. Furthermore, in the 21st century, many populations of small mammals may undergo a transformation to a noncyclic dynamic [59].

## 5. Conclusions

Individuals with extreme body condition indices represent a negligible proportion (1.33% with a BCI < 2 and 0.52% with a BCI > 5) of processed small mammals.

We believe that body condition in general is related to the use of high-calorie foods. Among underfit individuals (BCI < 2), the proportion of insectivores and herbivores was higher, while that of granivores and omnivores was less than expected. Among overfit individuals (BCI > 5), the proportion of granivores and insectivores was higher, while that of omnivores and herbivores was less than expected. *M. minutus* was characterized by the absence of underfit and highest proportion of overfit individuals.

The demographic structure (sex composition and age distribution) of individuals with extreme body condition indices was not homogeneous or well expressed. With the exception of *S. araneus*, the difference in the male-to-female ratio was not reliable in either the best- or the worst-fit individuals. Among underfit individuals, the number of juveniles was higher than expected in *S. araneus*, *S. minutus*, and *C. glareolus*, while the number of subadults was overrepresented in M. *arvalis*. The number of adults was disproportionately represented in *A. oeconomus*. Among overfit individuals, juveniles of *M. musculus* and *A. agrarius* were overrepresented, while the proportion of adults in both species was lower than expected.

The discovery that individuals with the highest body mass were not among those with the highest BCI contributes to the interpretation of the Chitty effect. For the first time in mid-latitudes, we report individuals of very high body mass in three shrew species.

## Figures and Tables

**Figure 1 life-14-01028-f001:**
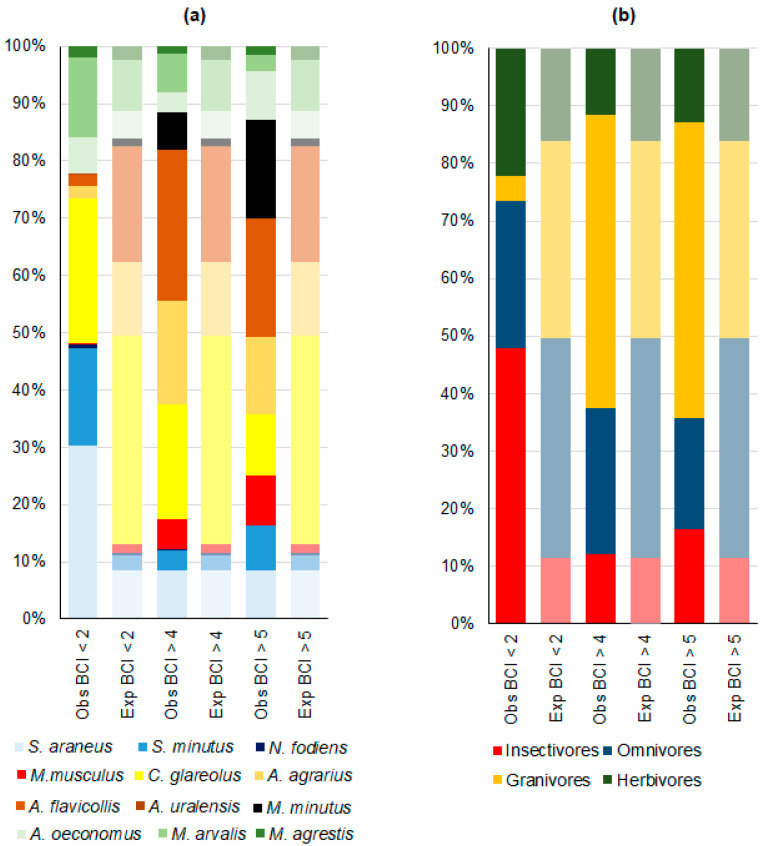
The observed (Obs) and expected (Exp) extreme BCI values for small mammal species (**a**) trapped in Lithuania between 1980 and 2023, and their trophic groups (**b**). The expected distribution (paler colors) was calculated to correspond to the number of individuals in the sample.

**Figure 2 life-14-01028-f002:**
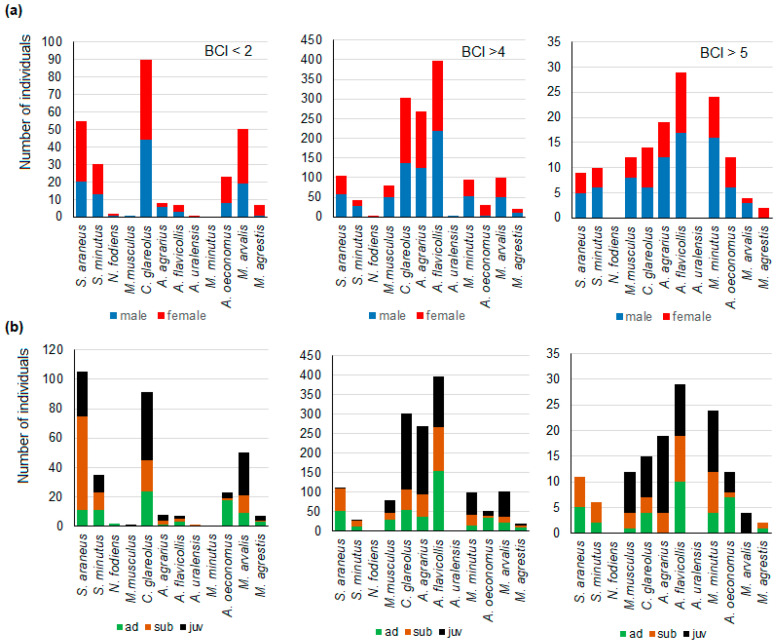
The distribution of individuals with extreme BCI values in small mammals according to sex (**a**) and age (**b**).

**Table 1 life-14-01028-t001:** Sample composition of small mammals used for this study.

Trophic Group	Species	*N*	%
Insectivores	Common shrew (*Sorex araneus*)	2303	8.5
	Pygmy shrew (*Sorex minutus*)	724	2.7
	Water shrew (*Neomys fodiens*)	99	0.4
Omnivores	House mouse (*Mus musculus*)	424	1.6
	Bank vole (*Clethrionomys glareolus*)	9866	36.4
Granivores	Striped field mouse (*Apodemus agrarius*)	3482	12.9
	Yellow-necked mouse (*Apodemus flavicollis*)	5403	20.0
	Pygmy field mouse (*Apodemus uralensis*) *	68	0.3
	Harvest mouse (*Micromys minutus*)	337	1.2
Herbivores	Root vole (*Alexandromys oeconomus*)	1286	4.8
	Common vole (*Microtus arvalis*) **	2429	9.0
	Short-tailed vole (*Microtus agrestis*)	652	2.4

* Species present only in N, NW, and NE parts of Lithuania since 1993. *** Sensu lato.* In most studies, the sibling vole, *Microtus rossiaemeridionalis*, was not specifically identified.

**Table 2 life-14-01028-t002:** The distribution of extreme BCI values in small mammals trapped in Lithuania from 1980 to 2023. Proportions significantly above the average are marked in gold, while those below the average are marked in blue.

Trophic Group	Species	*N*	BCI < 2		BCI > 4		BCI > 5	
			*n*	%	*n*	%	*n*	%
Insectivores	*Sorex araneus*	2303	109	4.73	126	5.47	12	0.52
	*Sorex minutus*	724	62	8.56	54	7.46	11	1.52
	*Neomys fodiens*	99	2	2.02	2	2.02	0	0.00
Omnivores	*Mus musculus*	424	1	0.24	79	18.63	12	2.83
	*Clethrionomys glareolus*	9866	91	0.92	303	3.07	15	0.15
Granivores	*Apodemus agrarius*	3482	8	0.23	270	7.75	19	0.55
	*Apodemus flavicollis*	5403	7	0.13	397	7.35	29	0.54
	*Apodemus uralensis*	68	1	1.47	1	1.47	0	0.00
	*Micromys minutus*	337	0	0.00	96	28.49	24	7.12
Herbivores	*Alexandromys oeconomus*	1286	23	1.79	53	4.12	12	0.93
	*Microtus arvalis*	2429	50	2.06	101	4.16	4	0.16
	*Microtus agrestis*	652	7	1.07	20	3.07	2	0.31
Total		27,073	361	1.33	1502	5.55	140	0.52

**Table 3 life-14-01028-t003:** Distribution of extreme BCI values in adult small mammals in relation to the intensity of reproduction (RI). F For males, 1+, 2+, and 3+ represent the intensity of spermatogenesis, while for females, 1, 2, and 3 represent the number of litters.

Sex	RI	*N*	BCI	*n*	*S. araneus*	*S. minutus*	*M. musculus*	*C. glareolus*	*A. agrarius*	*A. flavicollis*	*M. minutis*	*A. oeconomus*	*M. arvalis*	*M. agrestis*
Male	1+	500	<2	3				1				2		
			>4	19	1	3	8	1		3	1	1	1	
			>5	2						1	1			
	2+	410	<2	0										
			>4	17			3	1	3	6		1	1	2
			>5	0										
	3+	857	<2	3										
			>4	9			2	3	2	2				
			>5	0										
Female	1	1078	<2	10				5	1	1		1		2
			>4	52		1		7	10	20	3	10	1	
			>5	4						2		2		
	2	416	<2	8				5		1			2	
			>4	1						1				
			>5	0										
	3	64	<2	0										
			>4	1						1				
			>5	0										

**Table 4 life-14-01028-t004:** Statistics of the body mass (BM, g) for individuals with a BCI > 5.

Species	Juveniles			Subadults			Adults		
	*n*	Mean ± SE	BM Range	*n*	Mean ± SE	BM Range	*n*	Mean ± SE	BM Range
*S. araneus*				6	7.33 ± 0.33	6.0–8.5	6	9.65 ± 1.19	7.5–13.7
*S. minutus*				4	3.13 ± 0.24	2.5–3.5	2	4.82 ± 0.85	4.0–5.7
*M. musculus*	8	8.06 ± 0.71	5.5–11.5	3	15.28 ± 0.71	13.8–16.0	1	13.8	
*C. glareolus*	8	13.3 ± 0.90	10.1–17.0	3	19.77 ± 1.13	18.3–22.0	4	23.8 ± 2.02	19–28.8
*A. agrarius*	15	14.02 ± 0.57	8.8–16.5	4	21.33 ± 1.42	19.0–24.8			
*A. flavicollis*	10	20.59 ± 1.88	11.4–28.3	9	31.37 ± 1.63	26.5–40.0	10	39.77 ± 3.79	18.1–60.0
*M. minutus*	12	6.09 ± 0.27	4.0–7.5	8	7.44 ± 0.36	6.0–9.5	4	10.1 ± 1.42	7.0–13.3
*A. oeconomus*	4	13.08 ± 2.67	7.2–18.5				7	37.33 ± 3.67	19.0–46.1
*M. arvalis*	4	14.3 ± 1.20	12.0–17.5						
*M. agrestis*				2	23.0		1	47.0	

**Table 5 life-14-01028-t005:** The threshold and maximum body mass (BM, g) of adult small mammals and the statistics of the body condition index (BCI). In pregnant females, the embryo weight is excluded.

Species	Heaviest Adults						All Adults		
	BM Threshold	*n*	%	BCI ± SE	BCI Range	BM Maximum	*n*	BCI ± SE	BCI Range
*S. araneus*	14.0	12	1.9	3.35 ± 0.17 *	2.68–4.72	15.5	642	3.03 ± 0.03	1.15–6.25
*S. minutus*	5.0	5	3.2	3.98 ± 0.42 *	2.94–5.09	6.7	154	2.87 ± 0.06	1.33–5.72
*N. fodiens*	18.0	3	6.3	3.57 ± 0.49	3.02–4.54	19.1	48	2.72 ± 0.07	1.87–4.54
*M. musculus*	25.0	5	2.7	3.29 ± 0.13	2.77–3.45	30.3	183	3.47 ± 0.04	2.24–5.03
*C. glareolus*	35.0	13	0.4	3.46 ± 0.13 **	2.4–4.14	47.5	3367	2.84 ± 0.01	1.67–5.47
*A. agrarius*	35.0	15	2.1	3.29 ± 0.17	2.25–4.57	44.9	726	3.02 ± 0.02	1.85–4.97
*A. flavicollis*	60.0	13	0.5	3.36 ± 0.14	2.81–4.31	66.5	2548	3.19 ± 0.01	1.54–6.89
*A. uralensis*	30	1		2.40		32.0	19	2.58 ± 0.06	2.14–2.96
*M. minutus*	12.0	4	6.9	4.52 ± 0.30 *	3.68–5.00	14.0	58	3.63 ± 0.09	2.28–6.21
*A. oeconomus*	65.0	10	1.5	2.90 ± 0.11	2.31–3.50	77.0	666	2.89 ± 0.02	1.51–6.38
*M. arvalis*	45	5	0.7	3.34 ± 0.15 *	2.82–3.64	51.5	725	2.89 ± 0.02	1.49–4.95
*M. agrestis*	45.0	9	3.0	3.40 ± 0.35	2.66–6.10	56.0	301	2.88 ± 0.03	1.84–6.10

* Differences significant at *p* < 0.05; ** differences significant at *p* < 0.01.

## Data Availability

This is ongoing research; therefore, data are available from the corresponding author upon request.

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
