# Peer review of "Extreme Body Condition Index Values in Small Mammals"

_life, 2024, doi:10.3390/life14081028_

Round 1
Reviewer 1 Report
Comments and Suggestions for Authors
To improve the paper and its conclusions, the following questions, which need to be answered in the main text, are addressed:
- Given that the number of individuals analyzed varies significantly among species, are the samples representative of the species' range and abundance in the country?
- Were all individuals frozen upon capture? Why was the weight not assessed immediately after capture? Immediate weight assessment would be more accurate.
- Is the ratio of juveniles to adults similar across all species?
- The samples analyzed were obtained within a period of 43 years. Is this period verified in how many species?
· In the above context are there reported differences in average annual temperature in Lithuania from 1980 to 2023? This is relevant due to the reported association between temperature and body size in mammals
- Are there distinct climatic identified regions in Lithuania?
Author Response
Rev#1 comments and answers
Comment: Given that the number of individuals analyzed varies significantly among species, are the samples representative of the species' range and abundance in the country?
Answer: short answer is “yes”, except one species. Sample size of 11 species is representative of their numbers, and all 11 species are distributed across all Lithuania. Apodemus uralensis is present only in N, NW, and NE parts of Lithuania since 1993. Explanation added as footnote to Table 1.
Comment: Were all individuals frozen upon capture? Why was the weight not assessed immediately after capture? Immediate weight assessment would be more accurate.
Answer: we added few words about this. In fact, in 1970s-2000s, nearly all material was processed immediately (in the same day). However, later it was not always possible due to different reasons of logistics (trapping sites had no necessary facilities, or obtained number of individuals was too big). In this case, individuals were stored in (separate) plastic bags in frozen state, and processed ASAP.
Comment: Is the ratio of juveniles to adults similar across all species?
Answer: Short answer is “no”, but please note – when calculating expected frequencies, we employ observed proportions, so the question is not critical.
To acknowledge comment, we put reference to original Table 1 in our paper [8], presenting numbers, and add Appendix Table A1 to show proportions. Hope you will accept such approach.
Comment: The samples analyzed were obtained within a period of 43 years. Is this period verified in how many species?
Answer: out of 12 analyzed species, only one, Apodemus uralensis was not found during all investigation period. This was new species in Lithuania, present prom 1993. Text added as footnote to Table 1.
Comment: In the above context are there reported differences in average annual temperature in Lithuania from 1980 to 2023? This is relevant due to the reported association between temperature and body size in mammals
Answer: From 1994 to 2020 the mean annual air temperature increased by 0.037°C per year, according to Augustaitis, A., et al. (2022). Integrated effect of environmental changes on forest ecosystems in Lithuania: Strategies for adaptation to and mitigation of the main threats of global climate change. Eurasian Journal of Forest Research, 22, 45-48.
We, however, intend to analyze temporal trends of BCI in separate publication. As it was shown in Balčiauskas, L., & Balčiauskienė, L. (2024). Insight into Body Condition Variability in Small Mammals. Animals, 14(11), 1686, decade was not a strong, but significant factor of the BCI variability. Therefore, we do not add these data to the current manuscript.
Comment: Are there distinct climatic identified regions in Lithuania?
Answer: No, Lithuania does not have distinctly separate climatic zones. The climate of Lithuania is generally classified as humid continental, with some regional variations influenced by proximity to the Baltic Sea. In fact, country is flat and too small to have distinct zones. Explanation inserted in the beginning of Study site description..
Reviewer 2 Report
Comments and Suggestions for Authors
Comments on the manuscript “Am I too fat? Extreme body condition index values in small mammals” submitted to the Life
The manuscript is a compilation and analysis of BCI of 12 species of small mammals in Lithuania over 43 years. The aim was to analyze extreme BCI and body mass as a proxy, grouping animals by sex, age group, reproductive status, species, and trophic niche. A total of 27,073 individuals captured by snap trap were analyzed. To better understand the trophic niches and ecological pressures, the 12 species were grouped as omnivores, granivores, herbivores, and insectivores. There was little representation in extreme BCI, emphasizing the low representation of Chitty effect. The authors argue that body condition, in general, is related to the use of high-calorie foods.
Substantial data on the small mammal population in Lithuania over 43 years were collected. The data allow for a broad analysis of BCI, body mass, and other parameters (not shown) of the small mammal population.
The title is eye-catching and may catch the reader's attention, but it does not correspond to what is presented in the manuscript. The analysis method addresses extreme low or high BCI. It is not just about “too much fat”. The sentence “Am I too fat?” does not correspond to what was studied, despite the figurative speech effect.
In the introduction, the justification for the study tries to support itself with many arguments, but the focus of the study's importance is lost. For example, Why comment on pollution and BCI (line 66)? The study focuses on climate change and latitude to justify BCI variations.
The methods section is the weakest part of the manuscript because it is not sufficiently described. I begin by asking for more details about the region (country) where the study was conducted. Lithuania is well known to Europeans from high latitudes, but some scientific community knows little about this country's ecological conditions, climate, etc. Even so, the BCI of small mammals from Lithuania is compared with the BCI of small mammals from other countries or regions, such as Poland, Germany, and the USA. To compare with their findings, authors often present data from large countries, such as the USA, which have several ecosystems and many populations at many latitudes, making comparisons difficult. Other times, studies are cited from regions (e.g., Middle Yenisey, line 308) that may be well known to the authors but which will hardly be widely known to readers if they are not located in which countries, latitudes, or other geographic indicators. This comparison is imprecise and does little to validate regional comparisons of BCI of small mammals from other investigations with Lithuania.
There are significant gaps when describing the methods, starting from line 118. Individuals can vary in size depending on the season, the year (shortages vs. abundance), altitudes, etc. A temporal approach is essential for the reader to understand the variations in mass and BCI. The capture periods are not shown, so it isn't easy to think of a general approach, mixing very different periods of food availability (see, for example, Nieminen et al., 2015; Soledade Lemos et al., 2020; Ross et al., 2021).
The authors applied the body condition index (BCI) (line 153) based on the formula proposed by Moors (1985): BCI = (Q/L3) × 105 [53]. This formula may have been helpful in Moors’ (1985) study of Rattus norvegicus, but it has many limitations, as pointed out by several authors (Peig & Green, 2009; Wishart et al., 2023). The authors do not self-criticize the method's limitations when employing BCI in four taxonomic groups that occupy very different trophic niches and life histories and have very different ecology pressures.
Some arbitrary definitions are used in research, but this study is a quantitative approach, requiring some logical explanation for the operational decision of “BCI < 2.0 and BCI > 4.0, with BCI > 5.0 considered exceptionally large” (line 160). The Chitty effect is not a mere arbitrary decision. For example, in the study of Microtus califomicus, Lidicker, and Ostfeld (1991) did not make merely arbitrary determinations to define the weight > 55 g could reasonably be considered extra-large individuals (see page 109 from the authors' article). Boonstra and Krebs (1979) postulate that the ``Chitty effect'' is observed in cyclic populations of voles and lemmings is phase-related changes in the average body mass, with adults in high-density phase being 20% to 30% heavier than adults in the low-density phase of a cycle. Therefore, the arbitrary decision, without a quantitative and statistical basis, considering metabolic, ecological, physiological, and reproductive aspects, does not contribute to a reliable analysis in the present study of small mammals in Lithuania. As defined, the authors' arbitrary decision on BCI extremes is a mere assumption. In a quantitative study, a definition based on arbitrary assumptions is not enough when there are other ways to estimate the extremes of BCI better.
Why are there two extreme upper limits to define the Chitty effect (BCI>4 and BCI>5)? What does the definition of two limits contribute to the analysis?
The presentation of the data is broad and detailed, but there are some excesses. Why do the authors define three limits for the presentation of the data in Table 2 (BCI<2, BCI>4, BCI>5) but present the data in six intervals in Figure 1? Incidentally, Figure 1A does not seem important because it is redundant and confuses the reader regarding the analyses considering the other tables.
Finally, there is the impression in the manuscript that there are no relevant and new contributions to the findings shown in the manuscript. An article was published with the same data as the present manuscript (Balčiauskas & Balčiauskienė, 2024). There seems to be a cut of data already published and analyzed, including pointing out the Chitty effect in common and pygmy shrews. The present manuscript does not seem to bring anything new about what has already been published. I have doubts whether this “remastered” approach is sufficient to consider a publication in the Life.
References
Balčiauskas, L., & Balčiauskienė, L. (2024). Insight into Body Condition Variability in Small Mammals. Animals, 14(11), 1686.
Boonstra, R., and Krebs, C. J. (1979). Variability of large- and small-sized adults in fluctuating vole populations, Ecology 60, 567-573
Lidicker. W. Z.. Jr. and Ostfeld. R. S. (1991). Extra-large hody size in California voles: causes and fitness consequences. Oikos 61: 108-121.
Nieminen, P., Huitu, O., Henttonen, H., Finnilä, M. A., Voutilainen, L., Itämies, J., ... & Mustonen, A. M. (2015). Physiological condition of bank voles (Myodes glareolus) during the increase and decline phases of the population cycle. Comparative Biochemistry and Physiology Part A: Molecular & Integrative Physiology, 187, 141-149.
Ross, J. G. B., Newman, C., Buesching, C. D., Connolly, E., Nakagawa, S., & Macdonald, D. W. (2021). A fat chance of survival: body condition provides life-history dependent buffering of environmental change in a wild mammal population. Climate Change Ecology, 2, 100022.
Soledade Lemos, L., Burnett, J. D., Chandler, T. E., Sumich, J. L., & Torres, L. G. (2020). Intra‐and inter‐annual variation in gray whale body condition on a foraging ground. Ecosphere, 11(4), e03094.
Wishart, A. E., Guerrero-Chacón, A. L., Smith, R., Hawkshaw, D. M., McAdam, A. G., Dantzer, B., ... & Lane, J. E. (2023). Interpretation of body condition index should be informed by natural history. bioRxiv, 2023-01.
Author Response
Please find answers attached

Round 2
Reviewer 1 Report
Comments and Suggestions for Authors
The authors have addressed all comments and concerns.
However, I still believe that the manuscript could be significantly improved if the authors included the effect of temperature in the results.
Author Response
Rev#1 comments and answers, round 2
Comment: The authors have addressed all comments and concerns. However, I still believe that the manuscript could be significantly improved if the authors included the effect of temperature in the results.
Answer: analysis of the long term changes, as we wrote in the answers to Rev#1 round 1, will be separate publication. Les us remind, that current aim was different: “The objective of this study was to examine the distribution of small mammals exhibiting either extremely high (referred to as the "Chitty effect") or low body mass with respect to their trophic group, species, sex, age, and reproductive intensity. Additionally, we sought to estimate the relationship between body condition index and the mass of individuals of different species required for the Chitty effect to occur.”
If you refer to Bergmann’s rule (within a species, individuals in colder climates tend to be larger than those in warmer climates), then we prefer to have BCIs from different latitudes for comparison. Otherwise, conclusions could be wrong, as the influence of temperatures works through physiological and ecological factors, making the relationship indirect rather than direct.
Even mentioning temperature rise of 0.037°C per year as the factor would rise more questions than we could analyze along with other factors, such as r trophic group, species, sex, age, and reproductive intensity.
So we might only add text “Although we do not exclude the possible influence of climate warming on body mass loss and BCI, the lack of data from different latitudes does not allow us to exclude the possible occurrence of Bergmann's rule [15], so the data are not analyzed in temperature context.”
Reviewer 2 Report
Comments and Suggestions for Authors
Comments on the manuscript “Extreme body condition index values in small mammals” submitted to Life R2
There were two primary concerns regarding the manuscript. The first was a collection of gaps in the methods. The second concern was the use of data and an approach that did not seem to contribute anything new to the zoology and biometrics of small mammals since a similar approach was published in another journal.
The arguments for both concerns were explained or refuted in detail, making the text more understandable. The new writing included information on the climatic and geographic aspects of the region where the specimens were captured and an unequivocal definition of the criteria for categorizing extreme BCI, supported by citations from other studies.
It is acceptable to use an extensive biometric database accumulated over several decades to identify new ecological nuances, sex categories, age, etc. Thus, the article in its current form can be read and cited if it provides a new perspective for readers.
Author Response
Rev#2 comments and answers, round 2
Comments on the manuscript “Extreme body condition index values in small mammals” submitted to Life R2
There were two primary concerns regarding the manuscript. The first was a collection of gaps in the methods. The second concern was the use of data and an approach that did not seem to contribute anything new to the zoology and biometrics of small mammals since a similar approach was published in another journal.
The arguments for both concerns were explained or refuted in detail, making the text more understandable. The new writing included information on the climatic and geographic aspects of the region where the specimens were captured and an unequivocal definition of the criteria for categorizing extreme BCI, supported by citations from other studies.
It is acceptable to use an extensive biometric database accumulated over several decades to identify new ecological nuances, sex categories, age, etc. Thus, the article in its current form can be read and cited if it provides a new perspective for readers.
Answer: thank you